# Efficient plasmon-hot electron conversion in Ag–CsPbBr$_3$ hybrid nanocrystals

Xinyu Huang[1], Hongbo Li[2,6], Chunfeng Zhang[1,3], Shijing Tan[4], Zhangzhang Chen[1], Lan Chen[1], Zhenda Lu[2], Xiaoyong Wang[1] & Min Xiao[1,3,5]

Hybrid metal/semiconductor nano-heterostructures with strong exciton-plasmon coupling have been proposed for applications in hot carrier optoelectronic devices. However, the performance of devices based on this concept has been limited by the poor efficiency of plasmon-hot electron conversion at the metal/semiconductor interface. Here, we report that the efficiency of interfacial hot excitation transfer can be substantially improved in hybrid metal semiconductor nano-heterostructures consisting of perovskite semiconductors. In Ag–CsPbBr$_3$ nanocrystals, both the plasmon-induced hot electron and the resonant energy transfer processes can occur on a time scale of less than 100 fs with quantum efficiencies of $50 \pm 18\%$ and $15 \pm 5\%$, respectively. The markedly high efficiency of hot electron transfer observed here can be ascribed to the increased metal/semiconductor coupling compared with those in conventional systems. These findings suggest that hybrid architectures of metal and perovskite semiconductors may be excellent candidates to achieve highly efficient plasmon-induced hot carrier devices.

[1] National Laboratory of Solid State Microstructures, School of Physics, and Collaborative Innovation Center of Advanced Microstructures, Nanjing University, 210093 Nanjing, China. [2] College of Engineering and Applied Sciences, Nanjing University, 210093 Nanjing, China. [3] Synergetic Innovation Center in Quantum Information and Quantum Physics, University of Science and Technology of China, 230026 Hefei, Anhui, China. [4] Hefei National Laboratory for Physical Sciences at the Microscale, and Department of Chemical Physics, University of Science and Technology of China, 230026 Hefei, Anhui, China. [5] Department of Physics, University of Arkansas, Fayetteville, AR 72701, USA. [6] Present address: Key Laboratory of Flexible Electronics (KLOFE) and Institute of Advanced Materials (IAM), Jiangsu National Synergetic Innovation Center for Advanced Materials (SICAM), Nanjing Tech University, 30 South Puzhu Road, 211816 Nanjing, China. These authors contributed equally: Xinyu Huang, Hongbo Li. Correspondence and requests for materials should be addressed to C.Z. (email: cfzhang@nju.edu.cn) or to Z.L. (email: luzhenda@nju.edu.cn) or to M.X. (email: mxiao@uark.edu)

Surface plasmon resonance (SPR) in metallic nanostructures can directly convert absorbed photons into electrical energy by generating highly energetic electrons, i.e., hot electrons[1–3]. Collecting the energy of hot electrons by contacting metallic nanostructures with molecules or semiconductors can be integrated in optoelectronic devices for photovoltaic[4–7], photodetection[8–11], and photocatalytic applications[4,7,12–21]. This new design paves a way to realize hot-carrier devices whose performance may potentially exceed those of conventional devices.

Heterostructures consisting of metal and semiconductor nanoparticles have been widely studied for plasmon-derived hot-electron devices (i.e., the metal–semiconductor Schottky junction devices)[8–11]. The performances of such devices strongly depend on the efficiencies of plasmon-hot electron conversion at the metal–semiconductor interfaces. In the past few years, such a plasmon-hot electron conversion scenario has been intensively

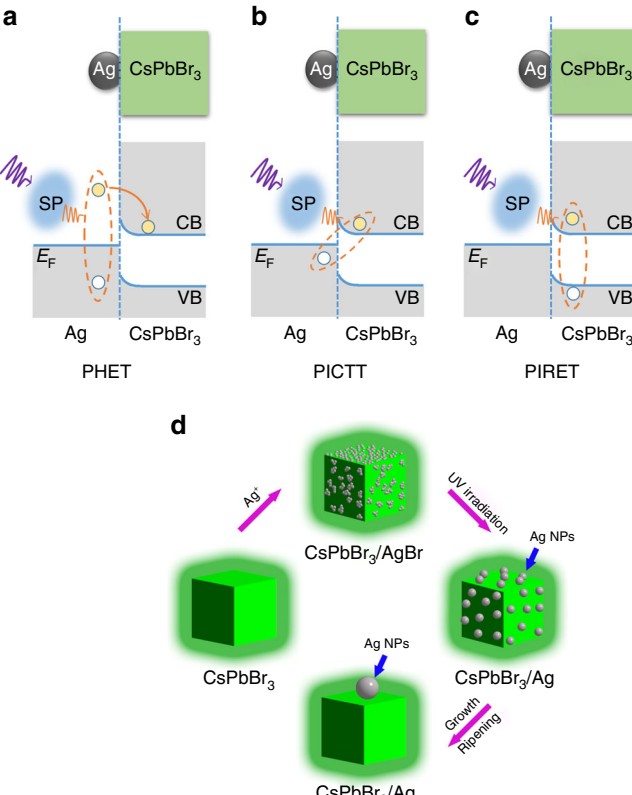

**Fig. 1** Scenario of interfacial excitation transfer processes. Energy alignment and possible pathways for plasmon-hot electron conversion in Ag–CsPbBr$_3$ hybrid nanocrystals (NCs). **a** Scheme of the conventional plasmon-induced hot-electron transfer process (PHET). A photoexcited plasmon (SP) decays into an electron–hole pair (solid and open circles in the dashed ellipsoid) in the Ag nanoparticle, followed by the transfer of hot electron into the conduction band (CB) of the CsPbBr$_3$ NC as indicated by the solid line with an arrowhead. **b** Scheme of the plasmon-induced charge-transfer transition process (PICTT). The plasmon decays by creating an electron in the CB of the CsPbBr$_3$ NC and a hole in the Ag nanoparticle. **c** Scheme of the plasmon-induced resonant energy transfer process (PIRET). The plasmon decays with simultaneous formation of an electron in the CB and a hole in the valence band (VB) of the CsPbBr$_3$ NC. The violet and orange wave lines indicate the optical excitation and Landau damping processes of plasmon in the Ag nanoparticle. The Fermi level of the Ag nanoparticle is labeled as $E_F$. **d** Proposed formation mechanism of Ag–CsPbBr$_3$ hybrid NCs. AgBr grown on the surface of CsPbBr$_3$ NCs decomposes to form small Ag nanoparticles under ultraviolet illumination whose sizes increase after prolonged irradiation via Ostwald ripening process

studied, for which distinct mechanisms have been proposed: the plasmon-induced hot-electron transfer (HET) from metal to semiconductor (Fig. 1a), the plasmon-induced charge-transfer transition (PICTT) across the interface (Fig. 1b) and the plasmon-induced resonant energy transfer (PIRET) that generate hot carriers in semiconductor directly (Fig. 1c)[1–4,14,22–27]. Conventionally, the plasmon-induced HET process is believed to occur following the Landau damping of surface plasmons into hot carriers in metal, which transfer across the interface[1,4,16,24]. The efficiency of plasmon-induced HET is limited by the ultrafast thermalization of hot electrons induced by electron–electron, electron–phonon scatterings, and the electron momentum conservation at the interface[1,26,28,29]. In contrast, Wu et al.[30] proposed the new scenario of PICTT to explain the high efficient HET in CdSe–Au nanorods, where the plasmons directly damp to form charge-separated electrons and holes. Direct transitions from metal to semiconductor have been reported in multiple metal/molecule systems[31–33], which can avoid the energy loss related to carrier thermalization. However, such a direct charge-transfer transition has weaker oscillation strength than that of an electronic transition in bulk metal or semiconductor themselves. In competition with HET at the metal–semiconductor interface, the PIRET has also been recognized as another efficient channel for plasmon-hot electron conversion[34]. Owing to plasmon–exciton interaction, plasmonic energy transfers to the semiconductor part and generates electron–hole pairs (EHPs) in semiconductor directly. Generally, a strong coupling at the metal–semiconductor interface can promote the plasmon-hot electron conversion, as evidenced by Tan et al.[3] with time-resolved two-photon photoemission spectroscopy.

In spite of these remarkable processes, the resultant efficiencies of hot-carrier devices remain far below expectations[1,4]. The oscillation strength of SPR in the metal is generally much higher than that in semiconductors like CdS and TiO$_2$[35]. In general, the density of states of donors are thus much higher than the available density of states of acceptors in nano-heterostructures studied so far, setting a bottleneck for further improvement of plasmon-hot electron conversion. Recently, perovskite semiconductors of lead halides have emerged as excellent material systems for optoelectronic applications because of their extremely strong light-matter interactions[36–41]. The absorption cross section per unit volume of CsPbX$_3$ (X = Cl, Br, or I) nanocrystals (NCs) is more than one-order magnitude larger than that in conventional chalcogenide II–VI semiconductor NCs (i.e., CdSe and CdS) with similar bandgaps[42,43]. The large oscillation strength of interband transition in perovskite semiconductor NCs may be ideal to promote the efficiency of hot electron conversion at metal/semiconductor interfaces. Inspiringly, good compatibility between perovskite semiconductor NCs and noble metals in forming nano-heterostructures have recently been established[44–47].

In this work, we study the dynamics of plasmon-hot electron conversion in a hybrid system of Ag–CsPbBr$_3$ NCs using ultrafast transient absorption (TA) spectroscopy. We clearly observe HET from Ag to CsPbBr$_3$ on an ultrafast timescale (<100 fs) when the hybrid system is pumped resonantly to the local mode of SPR. The spectral characteristics of charge-separated states evidenced the presence of HET process at the metal/semiconductor interfaces. In addition, photoluminescence (PL) excitation spectroscopic study indicates the plasmon-enhanced light emission from semiconductor, implying a considerable role of PIRET process with efficiency of ~15 ± 5% at the Ag/CsPbBr$_3$ interface. The efficient interfacial hot electron/energy transfer probably arises from the improved ratio between density of states of semiconductor acceptor and metallic donor in Ag–CsPbBr$_3$ NCs. The high efficiency of HET (~50 ± 18%) and the resultant

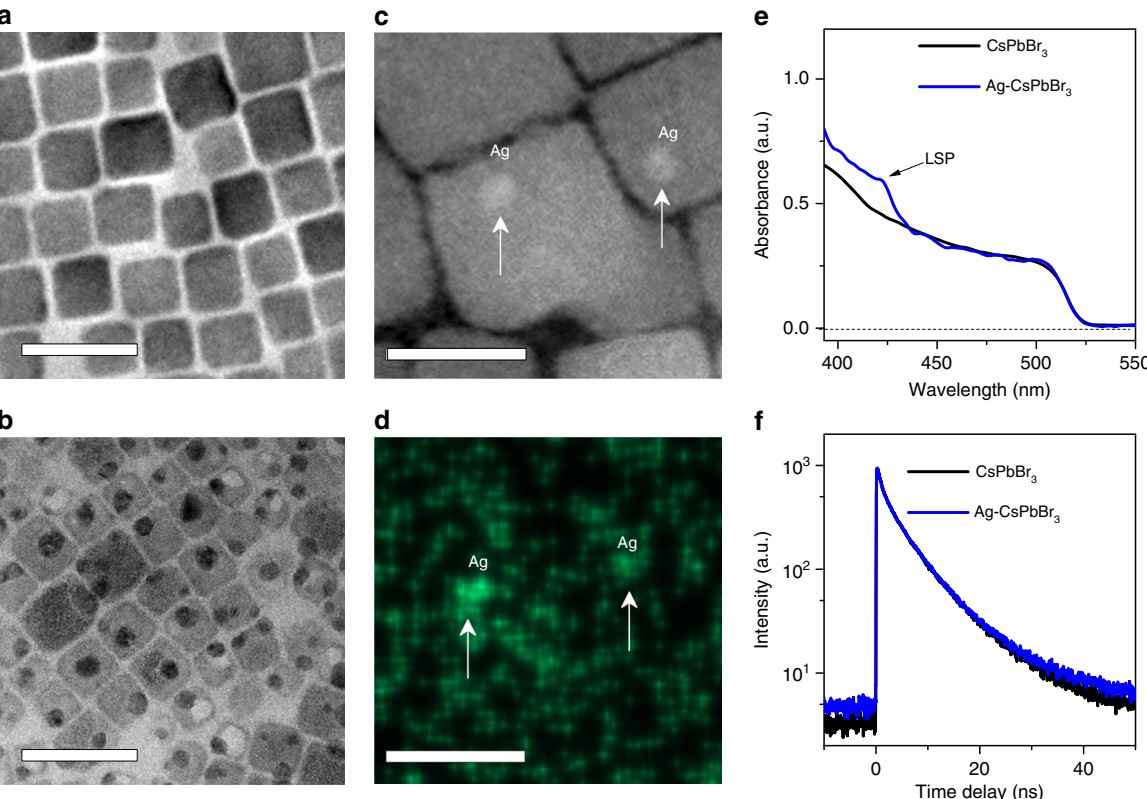

**Fig. 2** Characterizations of Ag–CsPbBr$_3$ nanocrystals (NCs). Typical TEM images of **a** neat CsPbBr$_3$ NCs and **b** Ag–CsPbBr$_3$ hybrid NCs (scale bar: 20 nm), respectively. **c** High-angle annular dark-field scanning TEM and **d** elemental mapping images for element Ag of Ag–CsPbBr$_3$ NCs at the same spot (scale bar: 10 nm). **e** Absorption spectra of neat CsPbBr$_3$ and Ag–CsPbBr$_3$ hybrid NCs. The absorption peak at ~424 nm is attributed to local mode of SPR. **f** Time-resolved PL traces recorded from neat CsPbBr$_3$ and Ag–CsPbBr$_3$ hybrid NCs, respectively

long-lived charge-separated states suggest a promising potential in applying the metal-perovskite semiconductor nanoheterostructures to further improve the performance of hot-carrier optoelectronic devices.

## Results

**Proximately coupled Ag–CsPbBr$_3$ NCs.** The possible pathways of plasmon-hot electron conversion in Ag–CsPbBr$_3$ hybrid NCs are schematically depicted in Fig. 1a–c. In principle, all the channels are beneficial for the strong coupling between metal and semiconductors. We establish the proximate coupling between Ag and CsPbBr$_3$ NCs by depositing Ag nanoparticles on the surfaces of semiconductor CsPbBr$_3$. We prepared the hybrid Ag–CsPbBr$_3$ NCs using a method inspired by a previous successful synthesis of Ag–TiO$_2$ hybrid systems[5]. Ag nanoparticles were quickly formed by a three-step procedure (Fig. 1d). When Ag ions are introduced, AgBr is grown on the surface of CsPbBr$_3$ NCs. Small Ag nanoparticles are subsequently formed during the decomposition of AgBr under ultraviolet illumination. These small Ag nanoparticles grow to a big one via Ostwald ripening process after prolonged ultraviolet irradiation. Interestingly, there is always one particular Ag nanoparticle that appears to be dominantly larger than the others deposited on each CsPbBr$_3$ NC, as shown in Fig. 2. The occurrence of such a ripening procedure is supported by the results of experiments under ultraviolet light with different exposure temporal durations (Supplementary Figure 1).

To confirm the structure of the Ag–CsPbBr$_3$ hybrid NCs, we carefully characterize the morphologies of neat CsPbBr$_3$ NCs and Ag–CsPbBr$_3$ NCs by transmission electron microscopy (TEM) as shown in Fig. 2a, b, respectively. The NCs have cubic shapes with average sizes of ~13 nm for both CsPbBr$_3$ and Ag–CsPbBr$_3$

hybrid NC samples. Dark dots are observed on the surfaces of cubic CsPbBr$_3$ NCs, which are the deposited Ag nanoparticles. These dark dots are highly reflective in scanning TEM images (Fig. 2c) as expected for the Ag metal, which is further confirmed by elemental mapping (Fig. 2d). X-ray photoelectron spectrum (Supplementary Figure 2) confirms the formation of Ag nanoparticles. In addition, signals from Ag$_2$O or residual AgBr are also observable. Ag$_2$O is possibly formed at the surface of Ag nanoparticles during sample preparation and X-ray spectroscopic measurements. The average diameter of the Ag nanoparticles is ~5.5 nm. The local mode of SPR in Ag–CsPbBr$_3$ hybrid NCs is manifested as an additional absorption peak with a full width at half maximum (FWHM) of ~130 meV at 424 nm (Fig. 2e), which is consistent with theoretical analysis (Supplementary Figure 3). This peak is absent from the absorption spectrum of the neat CsPbBr$_3$ NCs.

Time-resolved PL spectra are recorded to check the effect of SPR on the interband recombination dynamics. In principle, multiple factors may contribute to PL dynamics including the emission enhancement caused by passivation of surface traps and the fluorescence quenching of the semiconductor by the metal, which is also dependent on the size of metallic nanoparticles[45]. In this study, time-resolved PL spectra in Ag–CsPbBr$_3$ hybrid NCs with Ag nanoparticles of ~5.5 nm and neat CsPbBr$_3$ NCs are nearly the same at the late stage (Fig. 2f), implying that the presence of SPR has insignificant impact on the dynamics of interband electron–hole recombination of the CsPbBr$_3$ NCs at a long timescale (>200 ps).

**Interfacial carrier dynamics in Ag–CsPbBr$_3$ hybrid NCs.** We study the interfacial carrier dynamics in Ag–CsPbBr$_3$ hybrid NCs

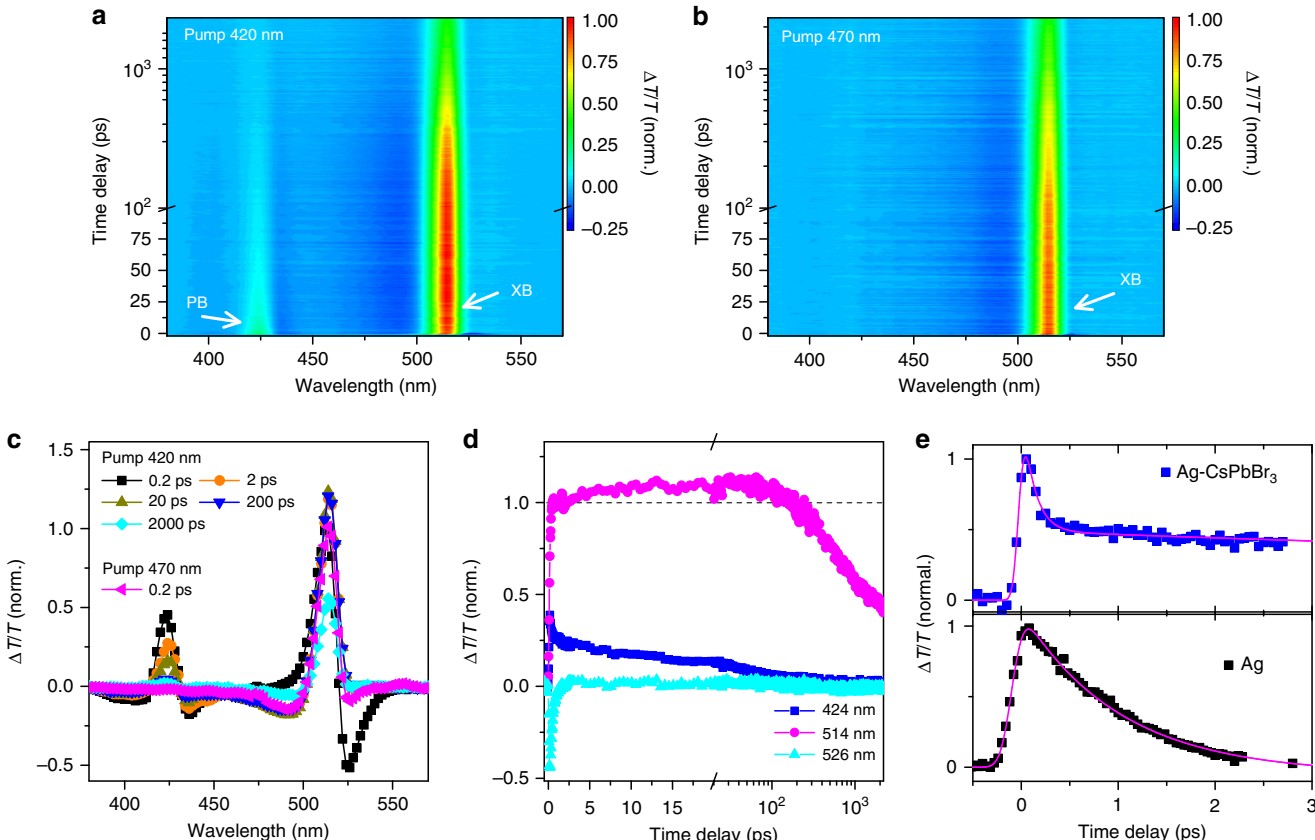

**Fig. 3** Carrier dynamics in Ag–CsPbBr$_3$ nanocrystals (NCs). TA data recorded from a solution sample of Ag–CsPbBr$_3$ NCs with pump wavelengths of **a** 420 and **b** 470 nm, respectively. **c** TA spectra recorded for Ag–CsPbBr$_3$ NCs at different time delays pumped at 420 nm. A TA spectrum recorded at a time delay of ~0.2 ps pumped at 470 nm is included as a reference. **d** TA dynamics of Ag–CsPbBr$_3$ NCs probed at different wavelengths. The pump wavelength is 420 nm. **a**–**d** The data are plotted in a scale normalized to the amplitude of signal probed at 514 nm (XB) at the delay of 0.2 ps. **e** Recombination dynamics probed at the wavelength of local SPR in the samples of neat Ag nanoparticles and hybrid Ag–CsPbBr$_3$ NCs. The solid lines are the curves fitted to the exponential decay function with a single component for the neat Ag nanoparticles and two components for the hybrid Ag–CsPbBr$_3$ NCs, respectively

by TA spectroscopy. To avoid the effect of many-body interaction, we have carefully checked power-dependent carrier dynamics (Supplementary Figure 4) and presented the data recorded under weak excitation with average number of absorbed photon per NCs <0.2 ($<N_{ex}><0.2$) (Supplementary Note 1). Figure 3a and b compare the TA spectra of a solution sample of Ag–CsPbBr$_3$ hybrid NCs recorded with a pump wavelength near the SPR at 420 nm and at an off-resonant wavelength of 470 nm, respectively. Under off-resonant pump, the TA data from Ag–CsPbBr$_3$ NCs (Fig. 3b and Supplementary Figure 5) are similar to the data recorded for the neat CsPbBr$_3$ NCs (Supplementary Figure 6) with a major bleaching band centered at 514 nm (denoted as XB), which can be naturally assigned to the state filling and bandgap renormalization[43,48]. In the weak pump regime, the signal amplitudes for both effects are proportional to the excited-state population near band edge. In addition to the XB signal, an additional bleaching signal emerges at 424 nm induced by the local SPR (denoted as PB) under resonant pump. The TA spectral feature for SPR is relatively narrower when compared with that of the neat Ag nanoparticles, which is plausibly caused by the strong coupling between Ag and CsPbBr$_3$. As an evidence, the TA feature for SPR in an overdoped sample becomes as broad as that in neat Ag nanoparticles (Supplementary Figure 7)[49].

We analyze the correlation between the dynamics of the PB and XB signals to illustrate the plasmon effect on the interfacial carrier dynamics. Figure 3c plots the TA spectra of the Ag–CsPbBr$_3$ NCs recorded under resonant pump at different

time delays, respectively. The amplitude of XB signal gradually increases in the first 20 ps upon resonant pump (Fig. 3c, d), which is not observed in the data recorded under off-resonant pump (Supplementary Figure 5). These results suggest the delayed rising behavior is triggered by the excitation of the local mode of SPR, implying the presence of plasmon-induced HET from Ag nanoparticles to CsPbBr$_3$ NCs.

The HET process is confirmed by the dynamics of photo-excited carriers in the Ag nanoparticles (Fig. 3e). We compare the dynamic traces probed at the wavelengths near SPR in hybrid Ag–CsPbBr$_3$ NCs and neat Ag nanoparticles (Fig. 3e). In neat Ag nanoparticles, the signal recovers exponentially with a lifetime parameter of ~0.8 ps (Fig. 3e), which is typical for hot-carrier thermalization as frequently observed in metallic structures (Supplementary Figure 8)[28,29,49,50]. For the hybrid NCs, the dynamic curve exhibits multiple exponential decay components. The lifetime of the major fastest component is <100 fs (as limited by the instrument) with an amplitude ratio of ~69%. The appearance of such an ultrafast decay component is evidence for hot-excitation transfer at the interface in Ag–CsPbBr$_3$ NCs that occurs on an ultrafast timescale that is faster than the thermalization of hot carriers in Ag nanoparticles. The TA spectra of Ag–CsPbBr$_3$ NCs show a slight shift to lower energy side near SPR in the first picosecond, which is likely to be caused by the charging effect of Ag nanoparticle as discussed in literatures[51,52]. The energy shift is insignificant due to low density of electron reduction[52].

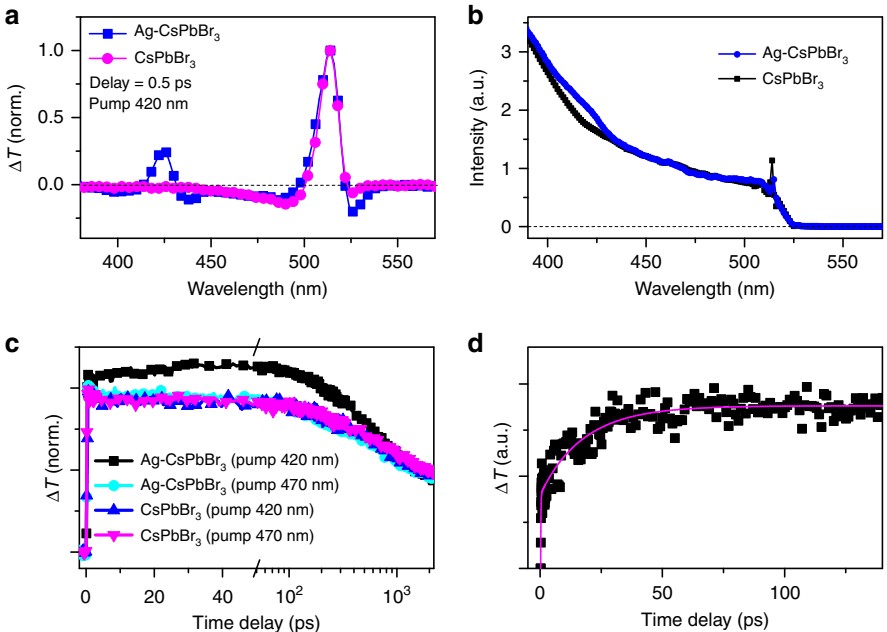

**Fig. 4** Hot-electron transfer (HET) dynamics in Ag–CsPbBr$_3$ nanocrystals (NCs). **a** TA spectra normalized to the signal amplitude at 514 nm recorded at a time delay of ~0.5 ps from CsPbBr$_3$ and Ag–CsPbBr$_3$ NCs, respectively. The pump wavelength is 420 nm. **b** PL excitation spectra recorded from CsPbBr$_3$ and Ag–CsPbBr$_3$ NCs. The spectra are normalized to the PL intensity under 470 nm excitation. **c** Kinetic curves probed at 514 nm recorded from CsPbBr$_3$ and Ag–CsPbBr$_3$ NCs pumped at the wavelengths of 420 and 470 nm, respectively. The curves are plotted normalized to the signal amplitude recorded at a delay of 2 ns. **d** Scaled kinetics of carrier transfer from Ag to CsPbBr$_3$ NCs in hybrid NCs obtained by subtracting the contribution of CsPbBr$_3$ from that of Ag–CsPbBr$_3$ NCs, whereas the signals for the two samples are normalized to the amplitude at a time delay of 2 ns. The solid line is the curve fitted to a biexponential growth function

Following the ultrafast decay component, the PB signal persists to a timescale of hundreds of picoseconds, which can be fitted with two additional decay components with lifetime parameters (amplitude ratios) of 12 ps (20%) and 150 ps (11%) (Supplementary Figure 9). These slow components are probably the consequence of formation of charge-separated states with excited electrons in the conduction band of CsPbBr$_3$ NCs and holes in silver nanoparticles. Such a charge separation process supports the presence of HET process at the Ag/CsPbBr$_3$ interface. The recombination of such a charge-separated state is much faster than the interband electron–hole recombination, which is possibly the reason for similar PL decay dynamics in CsPbBr$_3$ and Ag–CsPbBr$_3$ NCs. Remarkably, the lifetime is much longer than the thermalization of photoexcited carriers in Ag nanoparticles, indicating the successful conversion of short-lived plasmons in Ag nanoparticles into charge-transfer states with longer lifetimes. The lifetime extension of excited carriers is instrumental for improving the efficiency of charge dissociation, which is currently a major obstacle limiting the performance of plasmon-derived hot-carrier devices.

In addition to the correlated dynamics of XB and PB signals, the hot-excitation process should be also manifested in the short-lived photo-induced absorption band centered at 526 nm, which is known to be the excited-state absorption (ESA) from hot carriers[25,43]. The magnitude of the ESA signal under resonant pump is much larger than that recorded under off-resonant pump (Fig. 3c). Nevertheless, direct absorption of the shorter wavelength photons (420 nm) by CsPbBr$_3$ NCs may generate more hot carriers. To highlight the role of plasmon absorption in the hybrid NCs, the TA spectra of neat CsPbBr$_3$ NCs and hybrid Ag–CsPbBr$_3$ NCs pumped at 420 nm are compared in Fig. 4a. The experimental data clearly indicate that the ESA signal of hot carriers is markedly enhanced by the presence of local SPR in Ag–CsPbBr$_3$ NCs. Because the ESA signal of photoexcited

carriers in Ag nanoparticles is negligible at 526 nm (Supplementary Figure 8), the enhanced ESA signal can be naturally assigned to the local-plasmon-mode-derived HET from Ag nanoparticles to CsPbBr$_3$ NCs in the hybrid system.

The experimental observation of HET and charge separation at the Ag/CsPbBr$_3$ interface strongly supports the presence of HET process in the hybrid system. PIRET may co-exist in the same sample with strong plasmon–exciton coupling[34]. In such a resonant energy transfer process, EHP is created in semiconductor part at the cost of plasmon energy. Different from the formation of charge-separated states by HET process, PIRET leaves excitons in semiconductor part, which may recombine radiatively through the same channel for the excitons created directly by photoexcitation in semiconductor. We perform a comparative study on neat CsPbBr$_3$ and hybrid Ag–CsPbBr$_3$ NCs with PL excitation spectroscopy (Fig. 4b). The results clearly suggest the enhancement of PL emission when SPR is resonantly excited, suggesting a considerable role played by PIRET process.

**Quantification of hot-excitation transfer process**. We try to quantify the quantum efficiencies of different channels of hot-excitation transfer process in Ag–CsPbBr$_3$ NCs. In our experimental condition ($<N_{ex}><0.2$), the effect of multiple excitations in single particle can be neglected. In the hybrid system, the excited states of semiconductor may be formed with the EHP created by direct absorption, the EHP induced by PIRET, or the charge-separated state caused by HET. All three types contribute to the TA signal while the former two types involve in PL emission. It is reasonable to assume that the dynamics of EHP induced by PIRET is similar to that created directly by optical absorption in the semiconductor part. In this scenario, the efficiency of PIRET process can be evaluated to be ~15 ± 5% from the increment of PL emission (Fig. 4b).

To evaluate the quantum efficiency of HET process, we firstly extract the net TA signal related to the local SPR. The XB traces recorded from neat $CsPbBr_3$ and hybrid Ag–$CsPbBr_3$ NCs under resonant and off-resonant pumps are compared in a scale normalized to the signal at 2 ns in Fig. 4c, respectively. The major difference between the traces of the XB signals recorded under resonant (420 nm) and off-resonant (470 nm) pumps represent the contribution from plasmons. Under off-resonant pump, the TA signal is contributed by the EHP created by direct photoexcitation. Under resonant pump, the EHP created by PIRET and the charge-separated state populated by HET have also contributed to the signal. The charge-separated state has a much shorter lifetime in comparison to the interband electron–hole recombination (Fig. 3d). The late-stage dynamics is mainly contributed by the recombination of EHP, which is supported by the similar decay behaviors of the XB signals recorded from neat $CsPbBr_3$ NCs under different wavelength pump and hybrid Ag–$CsPbBr_3$ NCs under off-resonant pump (Fig. 4c). We subtract the normalized signals recorded under off-resonant pump from that with resonant pump to highlight the net dynamics of XB signals induced by plasmons (Fig. 4d). The faster component is typical for HET, while the slower component is observed in a timescale after the thermalization of hot carriers, which may be contributed by thermally activated electron transfer process[53], quantum tunneling process[54], and/or incoherent energy transfer[34]. To quantify the quantum efficiency of HET, we use a control sample of $CsPbBr_3$ NCs as reference (Supplementary Note 2) to minimize the potential inaccuracy caused by measuring the sample concentrations and pump intensities. In the normalized scale, it is estimated to be ~35 ± 15% increment of bleach signal at the delay of 50 ps in TA traces with pump near SPR in hybrid Ag–$CsPbBr_3$ NCs.

Because both EHP and charge-separated state population make contributions to the TA signal, it is essential to evaluate their amplitude ratio to extract the efficiency of HET. The major difference between the signal of an EHP and a charge-separated state is related to the different contribution between an electron and a hole in the TA signal. Such a difference is sensitive to the density of band-edge states, i.e., the degeneracy of band-edge states and the effective masses of electrons and holes. For the well-studied II–VI NCs like CdSe and CdS with wurtzite structure, the hole band has a much larger effective mass and a larger degeneracy than those of electron band, so the hole contribution to the XB signal is negligible in comparison with the electron contribution[16,55]. However, the contribution from holes may not be neglected for perovskite $CsPbBr_3$ NCs with cubic lattice structure. Theoretical calculations of band structures have shown band extrema at the R-points[56–58]. The density of states at the band edge contains contributions from both the halide and lead, and is complicated by the strong spin–orbit coupling. Previous theoretical studies have reported inconsistent results for the degeneracy of band-edge states[56–58]. Recently, a spectral survey on this issue by Klimov and co-workers suggested the equivalent degeneracy of electron and hole states at the band edge[59]. In addition, the effective masses of electron and hole bands are comparable. These results imply that the XB signal induced by each electron and hole are comparable in $CsPbBr_3$ NCs. That is, the signal amplitude of an EHP is twice of that of a charge-separated state population. The efficiency of HET can be estimated to be ~50 ± 18 % (Supplementary Note 2). The record high efficiency of HET in the heterodimer system of Ag–$CsPbBr_3$ hybrid NCs is about twice of magnitude larger than that measured in Au–CdS systems[30], suggesting a promising potential of hybrid metal/perovskite semiconductor system towards efficient hot-carrier technology.

## Discussion

Although many factors may be involved, the markedly high quantum efficiency of hot-excitation transfer in Ag–$CsPbBr_3$ NCs can be primarily ascribed to the strong coupling between metal and semiconductor. In previous plasmon-derived systems with metal–semiconductor or metal-molecule structures, the oscillation strength associated with local SPR is generally much higher than that of a single-electron transition in the semiconductor/molecule part, which limits the efficiency of hot-excitation transfer. In Ag–$CsPbBr_3$ NCs, the cross section of SPR absorption of Ag nanoparticles (5 nm) is $\sim 5 \times 10^{-14}$ $cm^2$, whereas the absorption cross section of $CsPbBr_3$ NCs is $\sim 10^{-13}$ $cm^2$ (refs. [42,60]). Such a configuration allows a higher density of accepting levels per donor, which is beneficial for achieving highly efficient HET and PIRET.

With strong coupling at metal/semiconductor interface, multiple channels of hot-excitation transfer may involve. We have connected the population of EHP and charge-separated state induced by plasmonic excitation to the PIRET and HET processes, respectively. As proposed by Li et al.[34], the PIRET caused by strong plasmon–exciton interaction converts plasmon energy to EHP in semiconductor prior to the loss of coherence. The presence of PIRET implies that the interfacial coupling in Ag–$CsPbBr_3$ NCs is possibly sufficient to generate hot carriers in semiconductor through plasmon damping. The conventional plasmon-induced HET process is predicted to be less effective (<10%) due to the restriction of linear momentum conservation of electrons[26]. The spectral overlap between metal and semiconductor in Ag–$CsPbBr_3$ NCs is favorable for the presence of PIRET[34]. The higher quantum efficiency of HET in Ag–$CsPbBr_3$ NCs observed here may be possibly caused by two effects: (1) The PICTT process as proposed by Wu et al.[25] may function here. The plasmon damping directly forms hot electrons in semiconductors at a high quantum efficiency. Unfortunately, because the timescale of plasmon damping (<10 fs) is beyond the temporal resolution of our measurement, we cannot explicitly distinguish the contributions from plasmon-induced HET or PICTT processes. Nevertheless, the strong coupling between Ag and $CsPbBr_3$ and the broad spectral linewidth of SPR (<10 fs) in Ag–$CsPbBr_3$ NCs implies the possibility of PICTT process. (2) The conservation of electron linear momentum is relaxed by the roughness of the junction[22]. Similar effect may also apply in the hybrid Ag–$CsPbBr_3$ NCs where the spherical Ag particles with average diameter of ~5 nm embed in the semiconductor NCs. A higher efficiency for HET may be expected if electron momentum conservation is relaxed with such a curved interface.

In general, PIRET and HET are competing channels for plasmon-derived hot-excitation transfer processes. Both channels are beneficial from the enhanced coupling between metal and semiconductor in Ag–$CsPbBr_3$ NCs. The coexistence of PIRET and HET processes is possibly related to sample heterogeneity. In some Ag–$CsPbBr_3$ NCs where residual AgBr at the Ag/$CsPbBr_3$ interface may disable the direct physical contact between metal and semiconductor[34], PIRET may dominate the process of hot-excitation transfer. In other Ag–$CsPbBr_3$ NCs with good metal–semiconductor contact, HET is likely to be the major channel responsible for the observed experiments. In principle, the process of hot-carrier transfer at metal/semiconductor interface can be enabled by either hot electrons or hot holes[61]. In the Ag–$CsPbBr_3$ NCs, the Fermi energy level of Ag is about 2.0 eV above the valence band of $CsPbBr_3$ NCs. However, charge separation has not been observed by hole transfer process when $CsPbBr_3$ NCs are selectively excited (Fig. 2b). Considering the energy band alignment of Ag and $CsPbBr_3$ NCs, hot electrons are the primary charges for hot-carrier transfer at the interface.

In summary, we have demonstrated highly efficient plasmon-hot electron conversion in Ag–CsPbBr$_3$ NCs as a model system of hybrid NCs consisting of metal and perovskite semiconductor. Efficient HET and PIRET processes have been observed with plasmon-induced EHP and charge-separated state population in CsPbBr$_3$ NCs, respectively. The processes occur on a timescale of <100 fs at the metal/semiconductor interface as evidenced by corroborating results including the post-excitation rising behavior in bleach signal of the CsPbBr$_3$ NCs, the acceleration of carrier recombination in Ag nanoparticles, the formation of a charge-separated state, and the enhancement of PL emission. The highly efficient hot-excitation transfer demonstrated here suggests that the hybrid architectures of metal and perovskite semiconductors may be excellent candidates to develop highly efficient plasmon-induced hot-carrier technology. The highly efficient plasmon-hot electron conversion can be applied to further improve the emerging technology of perovskite semiconductors based optoelectronic devices.

## Methods

**Sample preparation and characterization**. The samples of CsPbBr$_3$ NCs were synthesized following the approach developed by Kovalenko and co-workers[62]. The CsPbBr$_3$ NCs were dispersed into cyclohexane (~10 mg mL$^{-1}$). To incorporate Ag nanoparticles on the surface of CsPbBr$_3$ NCs, AgNO$_3$ (0.1 g) was added to TOP (10 mL), and then heated to 60 °C under stirring in a glove box. After the AgNO$_3$ dissolved completely, the solution was diluted with cyclohexane (100 mL) to form an Ag-TOP reagent. Then, CsPbBr$_3$ NC solution (12 mL) was added into 25 mL three-necked flask, and then heated to 50 °C. The Ag-TOP reagent prepared above was titrated into the three-necked flask at a rate of 0.2 mL min$^{-1}$ under UV light illumination for 5 min, and the continuously irradiated under ultraviolet light for up to 60 min to obtain the final products. Cyclohexane solutions of samples were added to 1 mm-thick quartz cuvettes with an optical density of ~0.25 at 500 nm for optical measurements. PL quantum yield in neat NCs is about 58 ± 5% while that in the sample of hybrid NCs we study is about 65 ± 6%. Neat Ag nanoparticles dispersed in cyclohexane with an average size of 6 nm were also prepared as reference samples according to the previous report[63].

The morphologies and compositions of the samples were characterized by high-resolution TEM (JEM-ARM200F) and high-resolution scanning TEM (a double-aberration corrected Titan™ cubed G2 60-300 S/TEM equipped with Super-X™ technology). Elemental mappings were acquired using a Super-X energy dispersive spectroscopy system composed of four silicon drift detectors covering a collection angle of 0.7 s rad to provide fast and efficient spectrum imaging collection. X-ray photoemission spectra were recorded for element analysis. X-ray photoelectron spectroscopy was conducted on a PHI 5000 Versa Probe delay line detector spectrometer equipped with a monochromatic Al Kα X-ray source.

**Optical measurements**. TA experiments were conducted using a commercial Ti: Sapphire regenerative amplifier (Libra, Coherent) at 800 nm with a repetition rate of 1 kHz and pulse duration of ~90 fs. An optical amplifier (OperA solo, Coherent) pumped by the regenerative amplifier was used to provide a pump beam with tunable wavelength. The probe supercontinuum source covering the spectral range from ultraviolet to red was generated by focusing a small portion of the femtosecond ultrashort pulses on a 5-mm CaF$_2$ plate. The plate was mounted on a moving stage to minimize the effect of laser damage. The TA signal was then analyzed by a high speed charge-coupled device (S7030-1006, Hamamatsu) with a monochromator (Acton 2358, Princeton Instrument) at 1 kHz enabled by a custom-built control board (Entwicklungsbüro Stresing). As reported previously[43,64], the system measures the signal of change in transmission ($\Delta T/T$) with the noise <5 × 10$^{-5}$ after averaging 500 couples of pump-on and pump-off spectra. The pump power was kept relatively low (~3 μJ cm$^{-2}$) with an average absorbed photon per dot <0.2 to minimize many-body effects (Supplementary Note 1). During TA measurements, the solution sample was stirred to suppress the photo-charging effect. The sample holder was placed in nitrogen atmosphere to avoid the potential sample damage related to oxygen and humidity. We checked the transmission spectrum with the supercontinuum before and after each round of TA measurements to confirm the stability of the sample (Supplementary Figure 10). For time-resolved PL measurement, a picosecond laser at 405 nm (LDH, Picoquant) was used as the excitation source. Time-resolved PL spectra were recorded by time-correlated single-photon counting using an avalanche photodiode with a temporal resolution of ~ 50 ps. For PL excitation measurement, light from a Xenon lamp is dispersed by a monochromator as the excitation light. PL intensity has been calibrated for the same excitation power at different wavelengths. All experiments were carried out at room temperature.

## Data availability

The experimental data that support the findings of this study are available from the corresponding authors upon reasonable request.

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

## Acknowledgements

This work was supported by the National Key R&D Program of China (2017YFA0303703 and 2018YFA0209101), the National Natural Science Foundation of China (21873047, 11574140, 91850105, 91833305, and 11621091), Jiangsu Provincial Funds for Distinguished Young Scientists (BK20160019), the Priority Academic Program Development of Jiangsu Higher Education Institutions (PAPD), and the Fundamental Research Funds for the Central Universities. C.Z. acknowledges financial support from the Tang Scholar Program. The authors acknowledge Dr. Xuewei Wu for his technical assistance and Dr. Weihua Zhang for stimulated discussion.

## Author contributions

C.Z., X.W., and M.X. conceived and designed the experiments. X.H. and L.C. performed the optical experiments. H.L., Z.C., and Z.L. prepared the samples. X.H., C.Z., S.T., and Z.L. analyzed the data. X.H. and C.Z. wrote the manuscript with help from all other authors.

## Additional information

**Competing interests:** The authors declare no competing interests.

