## [Peer Review File · Nature Communications]

Reviewers' comments:

Reviewer #1 (Remarks to the Author):

The manuscript reports the highly efficient plasmon-induced hot-electron transfer in hybridized Ag-CsPbBr₃ nanocrystals using transient absorption measurements. The observations in the work are interesting, and could motivate further researches on designing photovoltaic devices using metal and perovskite semiconductors, especially on improvement on plasmon-induced photoenergy conversions. As for very high quantum efficiency proposed in the manuscript, however, proposed quantity should be carefully discussed. This manuscript could be publishable, only after much more careful experiments with quantitative analysis.

1. For quantitative analysis on the states filling by electrons, dependence of the excitation light intensity is indispensable. The absolute intensity of transient absorption as well as relative components of lifetimes may depend on the light intensity. Comparison of these dependences between NCs with and without Ag particles could improve the accuracy on the estimation on the number of electrons occupying the conduction band states of NCs. Estimation on the number of excited electrons of Ag particles, based on the excitation light power and photon absorption (not scattering), could be also very important to compare the values obtained at NCs.

2. After electron injection due to effective charge separation, holes are remained in Ag nanoparticles. Reduction in the electron density of metal nanoparticles results in the shift of localized surface plasmon resonance to lower energy. It is not observed in the present system. Why? The color change in plasmons due to the difference in the number of electrons and holes in Ag nanoparticles could be also observed at the carrier relaxation process (long time scale), but not. The author should discuss carefully to understand the present charge separation process.

3. Ag nanoparticle synthesized in ambient condition is covered by Ag oxides with the band gap (or the HOMO-LUMO energy difference) with a few eV. Quality of Ag nanoparticles should be carefully checked by XPS or luminescent measurement to estimate the contribution of the photoexcitation by 420 nm.

4. Transient absorption spectrum of Ag nanoparticle plasmon at 424 nm is very sharp showing comparable full width half maximum of NCs. Excited plasmons generally show comparable transient features of broad width as previously reported (such as Phys. Rev. B 61, 6086 etc.) or as shown in Figure S2 for embedded Ag nanoparticle in NCs. This sharp feature may reflect the occupation of well-defined electronic states, not due to plasmon excitation.

5. The expression that "the slower component (18 ps) may be caused by hot electrons with energy slightly below the Schottky barrier" (Line 201-205) remains ambiguous for readers. The authors should explain physical origin much more carefully to understand the energy distribution of excited electrons at the interface.

Reviewer #2 (Remarks to the Author):

This manuscript describes the authors' work regarding plasmonic Ag nanoparticle decorated semiconductor perovskite CsPbBr₃ nanocubes and the resulting plasmonic energy transfer from the Ag nanoparticles to the CsPbBr₃ nanocubes. The transfer of energy from Ag nanoparticle to CsPbBr₃ nanocube was probed entirely using transient absorption (TA) spectroscopy. The authors claim that up to 70% efficiency for hot electron transfer (from Ag nanoparticle to CsPbBr₃) was achieved as determined by comparison of pristine CsPbBr₃ nanocubes and Ag decorated CsPbBr₃. The authors' attribute the extremely efficient transfer of the Ag nanoparticles' hot electrons to the high availability of conduction band-edge states available into which hot electrons can transfer. The authors' work has issues in interpretation of the collected data. As such, it is recommended that this manuscript be rejected for publications for the following reasons:

(1). The obtained hot electron transfer (HET) efficiency would exceed the maximum theoretical transfer efficiency for hot electrons from a plasmonic metal nanoparticle to a semiconductor of roughly 10%. This efficiency is limited by the kinetic energy and momentum of the interfacial hot

electrons normal to the semiconductor interface (see *J. App. Phys.*, 115 (2014), 134301). The designed structure is a typical plasmonic metal nanoparticle supported on semiconductor structure with no additional features to relax momentum restrictions for transferring hot electrons from Ag nanoparticle to CsPbBr₃. As such, it is extremely likely that there are issues with the interpretations of the obtained TA spectra and the subsequent HET efficiency calculations.

(2). Generally the HET can reach 70% only if chemical interface damping (CID) can occur (*Phys. Rev. B: Condens. Matter Mater. Phys.*, 1993, 48, 18178; *New J. Phys.*, 2003, 5, 151.; *ACS Nano*, 2017, 11, 2886–2893). The authors need to provide direct evidence to verify the possibility of CID process.

(3). The plasmonic energy transfer efficiency can achieve nearly 100% if a plasmon-induced resonance energy transfer (PIRET) process occurs instead of a hot electron transfer process. The authors rule out potential contributions due to PIRET from Ag nanoparticle to CsPbBr₃ due to the complete overlap between the Ag LSPR spectrum and CsPbBr₃ absorption (i.e. the LSPR energy overlap is beyond the bandgap energy) and the presence of TA signal contributions 100 – 1000 ps after the initial pump. The HET and PIRET process are competing processes that can both occur with the same plasmonic metal-semiconductor system on significantly different time-scales. PIRET would occur first within the first 50 femtoseconds (within the time delay before TA collection) before the plasmon loses its coherence while HET occurs well after the plasmon has lost its coherence on the order of 10-100 picoseconds later. It is suggested for the authors to read *Nature Photonics*, 9, 601–607 (2015) for better clarification on TA spectra when both HET and PIRET are possible. In addition, the complete overlap between LSPR and CsPbBr₃ absorbance spectrum does not rule out PIRET due to the potential for a super-bandgap transition from valence band to higher energy conduction band states during the dipole-dipole interaction.

Reviewer #3 (Remarks to the Author):

In this work, Huang et al describe Ag/CsPbBr₃ nanoparticle dimers, and use femtosecond uv/vis transient absorption to measure hot electron injection from the Ag nanoparticle to the CsPbBr₃ nanoparticle. They argue that electron transfer occurs in <100 fs with a quantum yield of ~70%, which is substantially greater than previous plasmon-semiconductor studies. This work is intriguing, but I have several conceptual concerns, as listed below. Furthermore, I'm not convinced of the impact of this work. In other studies of plasmon-induced carrier injection, the plasmonic enhancement extends the wavelength range in which the semiconductor absorbs, but in this case the plasmon is right on top of the semiconductor absorbance at ~420 nm. The Ag nanoparticle adds ~20% absorbance below 425 nm, so really they are at best getting 70% x 20% enhancement for a small region of the spectrum. Perhaps there is an argument to be made about selectively creating hot electrons vs hot holes, but they haven't made that argument.

The conceptual/interpretation concerns follow:

1. The long-time (recombination) behavior of neat CsPbBr₃ is identical to that of Ag/CsPbBr₃, as shown in Figure 4B and discussed on page 10, line ~193. It's not clear to me why this would be. In one case you have charge separation across an interface, and in the other case the electron and hole are confined to the perovskite nanocrystal. Why then would electron-hole recombination be the same in both cases? As noted below in the 'data presentation' questions, this makes me worry that the dimers are not surviving the photoexcitation conditions.
2. Similar to the last point, why should the photoluminescence decay be identical when you can get charge separation in Ag/CsPbBr₃ but not in CsPbBr₃? is the PL quantum yield the same in both, and if so why does that make sense?
3. The timescales of the claimed hot electron injection don't make sense with the yields. If hot electron injection is 100 fs, and the lifetime of the Ag nanoparticle itself is 800 fs, then the simple quantum yield branching would give an injection yield of 88% (and 100fs is the upper limit). They

measure a 69% amplitude ratio – why the loss in yield? I can imagine several possibilities (such as a particular energy threshold for hot electron injection), but they need to address all of these timescales in a more quantitative fashion.

4. The spectrum of perovskites at the band edge is a combination of bandgap renormalization and state-filling. Why is bandgap renormalization not considered when interpreting the signal near 525 nm?

5. The discussion of hot carrier multiplication on page 13 is purely speculative, with no experimental evidence. The carrier multiplication field is so fraught with arguments over experimental conditions, signal interpretation, etc, that such speculation should be treated very cautiously (especially given my concerns below about the reporting of experimental conditions)

In addition to these conceptual questions, several aspects of the data presentation are insufficient:

1. The pump power (in mJ/cm²) is never given for any experiment, nor is the OD of the sample. In addition, all experimental spectra are normalized, so the reader has no sense of the magnitude of the signals. Both are needed to accurately judge the presence of possible multiphoton effects (did they do the Poisson distribution calculation?)

2. The pump power and an accurate calculation of excitations/particle is especially important when trying to compare the Ag/CsPbBr₃ vs CsPbBr₃ spectra at early times. Nothing in the manuscript says that the Ag/CsPbBr₃ wasn't just pumped harder than the CsPbBr₃ sample, perhaps making more doubly-excited particles.

3. In Figure 3C, are the lines some kind of spline fit with the symbols showing the actual data? The symbols don't seem to line up vertically, which I don't understand. Since they use a monochromator and not an array CCD, I can't assume that they have regularly spaced data (especially since in their time traces they show each data point with a symbol). The lines in 3C seem too smooth to be the data (in which case the symbols are labels), but perhaps their instrument is in fact that good.

4. Spectra are frequently normalized without any description of how (at what data or time point). There are a couple places where this is explained (such as to the late-time signal in Figure 4B) but it needs to be explicit everywhere.

5. Do the Ag/CsPbBr₃ dimers survive the transient absorption experiment? There should be before/after UV/Vis and TEM studies.

6. What is the excitation wavelength for the TRPL studies? I didn't see this anywhere.

Response to the reviewers' comments for the manuscript # NCOMMS-18-02805:

We thank the reviewers for their insightful reviews of our manuscript. The critical comments/questions have helped us to rethink some of the problems and discussions in the original manuscript, and inspired us to perform several new measurements to better confirm our conclusions. In the following, we respond to the reviewers' comments point by point, and indicate changes made in the revised manuscript according to those comments. We hope that the reviewers will agree that the revised manuscript is much improved with their concerns fully addressed, so it is now suitable for publication on *Nature Communications*. The modifications in the manuscript have been highlighted by a different color.

Response to the comments by Reviewer #1:

General comment: The manuscript reports the highly efficient plasmon-induced hot-electron transfer in hybridized Ag-CsPbBr₃ nanocrystals using transient absorption measurements. The observations in the work are interesting, and could motivate further researches on designing photovoltaic devices using metal and perovskite semiconductors, especially on improvement on plasmon-induced photoenergy conversions. As for very high quantum efficiency proposed in the manuscript, however, proposed quantity should be carefully discussed. This manuscript could be publishable, only after much more careful experiments with quantitative analysis.

Our Response: We are grateful that the reviewer appreciates the significance of our work and recommends its publication after proper revisions. In the following, we answer the specific comments point by point.

Comment 1. For quantitative analysis on the states filling by electrons, dependence of the excitation light intensity is indispensable. The absolute intensity of transient absorption as well as relative components of lifetimes may depend on the light intensity. Comparison of these dependences between NCs with and without Ag particles could improve the accuracy on the estimation on the number of electrons occupying the conduction band states of NCs. Estimation on the number of excited electrons of Ag particles, based on the excitation light power and photon absorption (not scattering), could be also very important to compare the values obtained at NCs.

Our Response: In the original manuscript, we have reported the experimental data recorded under weak excitation. The average number of photons absorbed by a nanocrystal is ~ 0.2 . In this regime, the many-body effect is negligible since most of the excited nanocrystals absorb only one photon in the approximation of Poisson distribution. Following the reviewer's suggestion, we have included the data from power-dependent experiments (Figure R1). Under high pump power, the lifetime becomes shorter and the signal amplitude shows saturation, which are signatures of many-body effects (Figure R1). In addition, the power dependence of the signal amplitude in Ag-CsPbBr₃ nanocrystals shows a saturation behavior under a relatively

lower power excitation as compared to that in CsPbBr₃ nanocrystals (Figure R1C), which might be caused by the process of plasmon-induced resonance energy transfer (PIRET). To clarify this issue, we have included the power-dependent data (Figure R1) as new Supplementary Figure 4 and relevant discussions in SI (Supplementary Note 1). Following the models in literature (e.g., Govorov et al., J. Phys. Chem. C. 118, 7606 (2014), Forno et al., J. Phys. Chem. C 122, 8517–8527 (2018)), we roughly evaluate the number of hot charges per absorbed photon in the sample. The distribution of plasmon-induced hot electrons in ~ 5 nm metal nanoparticles is nearly flat from E_F to $E_F + \hbar\omega_{\text{plasmon}}$ (e.g., Govorov et al., J. Phys. Chem. C. 118, 7606 (2014), Forno et al., J. Phys. Chem. C 122, 8517–8527 (2018)). The Schottky barrier of Ag-CsPbBr₃ interface is estimated to be 0.6-0.7 eV, referred to semiconductors with similar energy band (Tung, Appl. Phys. Rev. 1, 011304 (2014)). The number of hot electrons with energy higher than the barrier is ~ 0.8 per photon absorbed by the nanoparticle. The efficiency of hot-electron transfer (HET) process is observed to be ~ 0.5 per photon absorbed by metal. Nevertheless, the efficiency of conventional HET is theoretically predicted to be inefficient if the conservation of linear electron momentum is considered (e.g., J. App. Phys. 115, 134301 (2014)). The highly efficient HET in this work is possibly related to the damping-induced hot electron generation in semiconductor part. The relevant discussion is included in Pages 14-16.

Figure R1. (A) Fluence-dependent kinetic dynamics of Ag-CsPbBr₃ hybrid nanocrystals probed at 514 nm. (B) The signal amplitudes probed at time delays of 1 ps and 3 ns are plotted versus pump fluence (average number of excitons per dot). (C) The power dependences of signal amplitudes at the delay of 3 ns for Ag-CsPbBr₃ and CsPbBr₃ nanocrystals are compared in a normalized scale.

Comment 2. After electron injection due to effective charge separation, holes are remained in Ag nanoparticles. Reduction in the electron density of metal nanoparticles results in the shift of localized surface plasmon resonance to lower energy. It is not observed in the present system. Why? The color change in plasmons

due to the difference in the number of electrons and holes in Ag nanoparticles could be also observed at the carrier relaxation process (long time scale), but not. The author should discuss carefully to understand the present charge separation process.

Our Response: We appreciate the reviewer's comment on the energy shift of LSPR with the reduction of electron density. We agree with the reviewer that charge separation process may principally charge the nanoparticles and induce an energy shift of LSPR like using the electrochemical treatments (e.g., Novo et al., JACS 131, 14664 (2009); Hoener et al., JPC Lett. 8, 2681 (2017)). We have carefully reexamined the experimental data and show the signal near plasmon resonance in Figure R2. A slight energy shift to the lower energy side is observed at the early stage (Figure R2B), which is possibly related to the energy shift of LSPR caused by the charging effect. Such an energy shift seems to recover at the late stage (Figure R2C). Nevertheless, the energy change of plasmon resonance is small, which can be ascribed to the weak excitation power. Under our experimental condition, less than one electron per particle is reduced through the photo-induced charge separation process. The energy shift in this regime was calculated to be insignificant (e.g., Herrera et al., Langmuir 32, 2829 (2016)), which well explains our observation. We have included additional discussion in the revised manuscript as “The TA spectra of Ag–CsPbBr₃ NCs show a slight shift to lower energy side near SPR in the first picosecond, which is likely to be caused by charging effect of Ag nanoparticle as discussed in literature. The energy shift is insignificant due to low density of electron reduction” on Page 9.

Figure R2. (A) TA data of hybrid Ag-CSPbBr₃ nanocrystals recorded in the spectral range of 400-450 nm under 420 nm pump. Normalized TA spectra at the time delays of 1.0 ps (B) and 200 ps (C) are compared with TA spectra recorded at the time delay of 0.1 ps, respectively.

Comment 3. Ag nanoparticle synthesized in ambient condition is covered by Ag oxides with the band gap (or the HOMO-LUMO energy difference) with a few eV. Quality of Ag nanoparticles should be carefully checked by XPS or luminescent measurement to estimate the contribution of the photoexcitation by 420 nm.

Our Response: Indeed, silver nanoparticles may be formed with Ag oxides. Following the reviewer's suggestion, we have measured the XPS spectrum of the

sample of Ag-CsPbBr₃ NCs and included it in the revised manuscript (Supplementary Figure 2, SI). The signal of silver oxide was observed in the XPS spectra with metal silver being the primary component. Silver oxide has a smaller gap with no peak feature near 420 nm in the absorption spectrum (Tjeng et al., Phys. Rev. B 41, 3190 (1990); Chen et al., App. Phys. Lett. 83, 5127 (2003)), so that its effect on the photoexcitation near 420 nm is insignificant. In addition, it is likely that a certain portion of oxidation is induced during XPS measurements since the Ag-CsPbBr₃ sample has almost no absorption at longer wavelength (> 550 nm). The data are included as Supplementary Figure 2 in SI with relevant discussion added on Page 7 in the revised manuscript.

Comment 4. Transient absorption spectrum of Ag nanoparticle plasmon at 424 nm is very sharp showing comparable full width half maximum of NCs. Excited plasmons generally show comparable transient features of broad width as previously reported (such as Phys. Rev. B 61, 6086 etc.) or as shown in Figure S2 for embedded Ag nanoparticle in NCs. This sharp feature may reflect the occupation of well-defined electronic states, not due to plasmon excitation.

Our Response: Indeed, the TA feature near LSPR is relatively narrow. The spectral feature is possibly caused by strong interaction between metal and semiconductor. Because of such strong coupling, the photo-excited plasmons in Ag-CsPbBr₃ nanocrystals damp in an ultrafast temporal scale that is beyond our measurement resolution. We have performed additional experiments to verify this argument. In a control sample with overdoped silver, the semiconductor compound is partially destructed (Figure R3A). We found that the linewidth (Figure R3) is as broad as the results reported in literature (Phys. Rev. B 61, 6086 (2000)). We include the data as Supplementary Figure 7 in SI and the relevant discussion as “*The TA spectral feature for SPR is relatively narrower when compared with that of the neat Ag nanoparticles, which is plausibly caused by the strong coupling between Ag and CsPbBr₃. As an evidence, the TA feature for SPR in an overdoped sample becomes as broad as that in neat Ag nanoparticles*” on Page 8 in the revised manuscript.

Figure R3. Carrier dynamics in an overdoped sample. (A) The absorption spectrum and (B) TA spectrum at the time delay of 0.2 ps of the overdoped sample. (C) The TA data of the overdoped sample are plotted versus time delay and probe wavelength.

Comment 5. The expression that "the slower component (18 ps) may be caused by hot electrons with energy slightly below the Schottky barrier" (Line 201-205) remains ambiguous for readers. The authors should explain physical origin much more carefully to understand the energy distribution of excited electrons at the interface.

Our Response: Following the reviewer's suggestion, we have explained the slower component more carefully in the revised manuscript. In the temporal scale of 18 ps, the plasmon already loses its coherence and damps into electrons. The relatively slower component may be caused by three processes: 1) The thermalized electrons transfer across the metal/semiconductor interface through a thermal activation process (Li et al., *J. Am. Chem. Soc.* **129**, 11535 (2007)); 2) Electrons with energy slightly below the Schottky barrier across the barrier through a quantum tunneling process (*Nature Nanotech.* **6**, 517 (2011)); 3) Plasmonic energy transfers incoherently from metal to semiconductor (Li et al., *Nature Photon.* **9**, 601 (2015)). All processes are slower as compared to the damping-induced interfacial excitation transfer. We have included a relevant discussion as "The faster component is typical for HET, while the slower component is observed in a timescale after the thermalization of hot carriers, which may be contributed by thermally activated electron transfer process, quantum tunneling process, and/or incoherent energy transfer" on Page 12 in the revised manuscript.

Response to the comments by Reviewer #2:

General comment: This manuscript describes the authors' work regarding plasmonic Ag nanoparticle decorated semiconductor perovskite CsPbBr₃ nanocubes and the

resulting plasmonic energy transfer from the Ag nanoparticles to the CsPbBr₃ nanocubes. The transfer of energy from Ag nanoparticle to CsPbBr₃ nanocube was probed entirely using transient absorption (TA) spectroscopy. The authors claim that up to 70% efficiency for hot electron transfer (from Ag nanoparticle to CsPbBr₃) was achieved as determined by comparison of pristine CsPbBr₃ nanocubes and Ag decorated CsPbBr₃. The authors' attribute the extremely efficient transfer of the Ag nanoparticles' hot electrons to the high availability of conduction band-edge states available into which hot electrons can transfer.

The authors' work has issues in interpretation of the collected data. As such, it is recommended that this manuscript be rejected for publications for the following reasons:

Response: We appreciate the reviewer's rigorous comments on our interpretation of the collected data. The insightful comments have stimulated us reconsidering some proposed models in the original manuscript. We have performed additional PL excitation (PLE) spectroscopic experiments and observed experimental evidences of plasmon-induced resonance energy transfer (PIRET) process. We agree with the reviewer that the PIRET process should be included for better interpretations of the experimental data. We have rewritten the discussion parts (Pages 14-16) and included the new experimental data (Figure 4B) in the revised manuscript. The specific comments are answered below.

Comment (1). The obtained hot electron transfer (HET) efficiency would exceed the maximum theoretical transfer efficiency for hot electrons from a plasmonic metal nanoparticle to a semiconductor of roughly 10%. This efficiency is limited by the kinetic energy and momentum of the interfacial hot electrons normal to the semiconductor interface (see J. App. Phys., 115 (2014), 134301). The designed structure is a typical plasmonic metal nanoparticle supported on semiconductor structure with no additional features to relax momentum restrictions for transferring hot electrons from Ag nanoparticle to CsPbBr₃. As such, it is extremely likely that there are issues with the interpretations of the obtained TA spectra and the subsequent HET efficiency calculations.

Our Response: We thank the reviewer for pointing out the paper in literature that predicted the efficiency limit of 10% (J. App. Phys., 115, 134301 (2014)). In that paper, the efficiency limit was evaluated in a planar junction of metal/semiconductor interface. It was predicted that the efficiency for hot electron transfer should be less than 10% due to the conservation of electron linear momentum in the planar junction. By including the contribution of the PIRET process, we revised the maximal efficiency (~ 50%) of hot-electron transfer. Nevertheless, the value is much higher than the limit predicted in literature (J. App. Phys. 115, 134301 (2014)). Two factors may be involved for the divergence between our experimental results and the theoretical prediction: 1) The limitation imposed by the conservation of electron linear momentum in the planar junction may be relaxed in the systems with rough metal/semiconductor interfaces (Giugni et al., Nature Nanotech. 8, 845 (2013)). If such conservation is relaxed by roughness of the junction, the probability of the

electron transfer may be significantly increased. Actually, several papers have reported the efficiency beyond the limit of 10% (e.g., Brongersma et al., *Nature Nanotech.* 10, 25 (2015); Furube et al. *JACS* 129, 14852 (2007); Ratchford et al., *Nano Lett.* 17, 6047 (2017)). In our work, spherical Ag particles with an average diameter of ~ 5.5 nm are embedded in the semiconductor nanocrystals. With such a curved interface, a higher efficiency can be expected with relaxed linear momentum conservation. 2) Highly efficient electron transfer is enabled by direct damping of plasmon with the formation of hot electrons in semiconductors (i.e., the second comment of the reviewer). Such a process may be involved in the hybrid nanocrystal samples we studied here (details are available in Response to comment 2 below). We have modified the discussion on this point (Pages 14-16) in the revised manuscript.

Comment (2). Generally the HET can reach 70% only if chemical interface damping (CID) can occur (Phys. Rev. B: Condens. Matter Mater. Phys., 1993, 48, 18178; New J. Phys., 2003, 5, 151.; ACS Nano, 2017, 11, 2886–2893). The authors need to provide direct evidence to verify the possibility of CID process.

Our Response: We appreciate the reviewer's suggestion. The broad absorption features imply the possibility of ultrafast damping (< 10 fs) that directly creates hot electrons in semiconductors (i.e., Wu et al., *Science* 349, 632 (2015); Tan et al., *Nature Photon.* 11, 806 (2017)). Unfortunately, due to the limitation of temporal resolution (~ 100 fs) in our measurement, we cannot resolve the exact dynamics of such a damping process. Alternatively, we have observed evidences of PIRET process by photoluminescence emission spectroscopy (Figure R4), indicating that the interfacial coupling is sufficiently strong to induce energy transfer prior to the loss of coherence (Li et al., *Nature Photon.* 9, 601 (2015)). In principle, such strong coupling between metal and semiconductor may cause the damping-induced electron transfer. We have clearly observed highly efficient charge separation induced by the hot electron transfer although the damping process cannot be distinguished exclusively due to the experimental limitation. We have included the new data as Figure 4B and discussed the possibility of such damping-induced electron transfer on Pages 14-15 in the revised manuscript.

Figure R4. Typical PL excitation spectra of neat CsPbBr₃ and hybrid Ag-CsPbBr₃

nanocrystals. The spectra are normalized to the PL intensity under 470 nm excitation.

Comment (3). The plasmonic energy transfer efficiency can achieve nearly 100% if a plasmon-induced resonance energy transfer (PIRET) process occurs instead of a hot electron transfer process. The authors rule out potential contributions due to PIRET from Ag nanoparticle to CsPbBr₃ due to the complete overlap between the Ag LSPR spectrum and CsPbBr₃ absorption (i.e. the LSPR energy overlap is beyond the bandgap energy) and the presence of TA signal contributions 100 – 1000 ps after the initial pump. The HET and PIRET process are competing processes that can both occur with the same plasmonic metal-semiconductor system on significantly different time-scales. PIRET would occur first within the first 50 femtoseconds (within the time delay before TA collection) before the plasmon loses its coherence while HET occurs well after the plasmon has lost its coherence on the order of 10-100 picoseconds later. It is suggested for the authors to read Nature Photonics, 9, 601–607 (2015) for better clarification on TA spectra when both HET and PIRET are possible. In addition, the complete overlap between LSPR and CsPbBr₃ absorbance spectrum does not rule out PIRET due to the potential for a super-bandgap transition from valence band to higher energy conduction band states during the dipole-dipole interaction.

Response: We thank the reviewer for reminding us the possible coexistence of PIRET and HET processes. We agree with the reviewer that the observation of charge separation states is a signature of electron transfer, which, however, cannot exclude the process of PIRET. We have carefully studied the paper noted by the reviewer (Li et al., Nature Photon. 9, 601–607 (2015)) and reanalyzed our experimental data. We have performed additional experiments and observed compelling evidence of PIRET process in the photoluminescence emission spectra. We observed ~ 15% enhancement of PL emission when the LSPR is excited (Figure R4), which can be regarded as a clear evidence for PIRET induced by the strong plasmon-exciton coupling. Some TA signals are possibly contributed by the PIRET processes. We carefully reanalyzed the measured data and the efficiency of electron transfer is estimated to be ~ 50% in our system (Supplementary Note 2). We have included the new data (Figure 4B) and made appropriate revisions on relevant discussions (Pages 14-16 and Supplementary Note 2) in the revised manuscript.

Response to comments by Reviewer #3:

General comments: In this work, Huang et al describe Ag/CsPbBr₃ nanoparticle dimers, and use femtosecond uv/vis transient absorption to measure hot electron injection from the Ag nanoparticle to the CsPbBr₃ nanoparticle. They argue that electron transfer occurs in <100 fs with a quantum yield of ~70%, which is substantially greater than previous plasmon-semiconductor studies. This work is intriguing, but I have several conceptual concerns, as listed below. Furthermore, I'm not convinced of the impact of this work. In other studies of plasmon-induced carrier injection, the plasmonic enhancement extends the wavelength range in which the

semiconductor absorbs, but in this case the plasmon is right on top of the semiconductor absorbance at ~420 nm. The Ag nanoparticle adds ~20% absorbance below 425 nm, so really they are at best getting 70% x 20% enhancement for a small region of the spectrum. Perhaps there is an argument to be made about selectively creating hot electrons vs hot holes, but they haven't made that argument.

Our Response: We are happy to see that the reviewer finds our work intriguing. We have included more data and experimental details to address the reviewer's conceptual concerns. In addition, we have strengthened the justification for the impact of this work. We agree with the reviewer that the sample of Ag-CsPbBr₃ nanocrystals can be regarded as a system that shows the interfacial transfer of plasmon-induced hot electrons. This work provides valuable knowledge towards resolving the argument noted by the reviewer. Of equal importance, our current work represents the first demonstration of efficient hot-excitation transfer in a hybrid system of metal and perovskite semiconductor nanostructures. Considering the high efficiencies of perovskite optoelectronic devices, further improvement in efficiency with plasmon effects as demonstrated in our work can be technically meaningful. More importantly, our work is a proof-of-principle study demonstrating the feasibility and superiority of using perovskite semiconductor nanostructures to harvest the plasmon energy, which will stimulate a rapidly-growing interest in developing hot-carrier technology using hybrid systems of metal/perovskite semiconductors with extended spectral coverage. We acknowledge the reviewer's suggestion of justifying the impact of this work with relevance to the "argument to be made about selectively creating hot electrons vs hot holes". We have included the above justifications in the revised manuscript as: "In principle, the process of hot carrier transfer at metal/semiconductor interface can be enabled by either hot electrons or hot holes. In the Ag-CsPbBr₃ NCs, the Fermi energy level of Ag is about 2.0 eV above the valence band of CsPbBr₃ NCs. However, charge separation has not been observed by hole transfer process when CsPbBr₃ NCs are selectively excited (Figure 2B). Considering the energy band alignment of Ag and CsPbBr₃ NCs, hot electrons are the primary charges for hot carrier transfer at the interface" on Paragraph 15-16 and "The highly efficient plasmon-hot electron conversion can be applied to further improve the emerging technology of perovskite semiconductors based optoelectronic devices" on Page 16. Below we answer the specific comments point by point.

The conceptual/interpretation concerns follow:

Comment 1. The long-time (recombination) behavior of neat CsPbBr₃ is identical to that of Ag/CsPbBr₃, as shown in Figure 4B and discussed on page 10, line ~193. It's not clear to me why this would be. In one case you have charge separation across an interface, and in the other case the electron and hole are confined to the perovskite nanocrystal. Why then would electron-hole recombination be the same in both cases? As noted below in the 'data presentation' questions, this makes me worry that the dimers are not surviving the photoexcitation conditions.

Response: We appreciate the reviewer's insightful comments. The comparable

dynamical behaviors of electron-hole recombination observed in the two samples at the late stage (> 1 ns) are understandable considering the short lifetime of charge separation state. As reported in the original manuscript and SI, the longest component of charge-transfer state has a lifetime of ~ 150 ps. Such a lifetime is much longer than the lifetime of hot electrons (sub-picosecond), which, however, is about one order of magnitude shorter than that of the interband electron-hole recombination in nanocrystals. Our experiment is in the weak excitation regime with an average number of ~ 0.2 photons absorbed per dot. Considering the Poisson distribution, most excited dots can have only one excited electron. In a hybrid nanocrystal, the excitation in CsPbBr₃ NCs may be formed by direct absorption, energy transfer or charge transfer processes. The electron hole pair created by either direct absorption or energy transfer recombines through an interband recombination channel like that in neat nanocrystals. The charge separate state induced by charge transfer recombines through an interfacial process with a shorter lifetime (< 150 ps). The above fact can probably explain the similar behaviors of electron-hole recombination at the late stage (> 1 ns) in the neat and hybrid nanocrystals but dramatic different dynamics at the early stage (< 150 ps). Some relevant discussions have been included as “*The recombination of such a CS state is much faster than the interband electron-hole recombination, which is possibly the reason for similar PL decay dynamics in CsPbBr₃ and Ag-CsPbBr₃ NCs at a late stage. Remarkably, the lifetime is much longer than the time scale for thermalization of photo-excited carriers in Ag nanoparticles, indicating the successful conversion of short-lived plasmons in Ag nanoparticles into charge-transfer states with longer lifetimes. The lifetime extension of excited carriers is instrumental for improving the efficiency of charge dissociation, which is currently a major obstacle limiting the performance of plasmon-derived hot-carrier devices*” on Page 9-10 of the revised manuscript.

Sample stability is an important issue that we have handled carefully during measurements. The solution sample was under continuous stirring to suppress the photocharging effect. The sample cell was placed in the nitrogen atmosphere to avoid the possible degradation due to moisture and oxygen. The transmission spectrum was monitored prior and after each scan of TA measurements using the probe light of white-light supercontinuum (see Figure R5 below). No obvious degradation was detected during a round of measurement (~ 3 hours). We have included the data as Supplementary Figure 10 to justify the photo stability of the sample.

Figure R5. The transmission spectra (A) and absorption spectra (B) of a solution sample before and after transient absorption measurements. The absorption spectra show clear consistency. The spectra were measured using supercontinuum as the light source. The slight disparity in the two curves is possibly due to the instability of the supercontinuum light source.

Comment 2. Similar to the last point, why should the photoluminescence decay be identical when you can get charge separation in Ag/CsPbBr₃ but not in CsPbBr₃? is the PL quantum yield the same in both, and if so why does that make sense?

Our Response: The underlying mechanism is similar to that for the above point. The charge separation state has a lifetime much shorter than that of the interband electron-hole recombination. Consequently, the charge separation has an insignificant effect on PL decay. Under the excitation at 405 nm, PL quantum yield in neat nanocrystals is about 58±5% while that in the sample of hybrid nanocrystals we study is about 65±6%. Further in-depth study is required to fully elucidate the mechanism of quantum yield of PL emission with metal deposition in CsPbBr₃ nanocrystals. Multiple factors may be involved such as the enhancement caused by passivation of surface traps and the fluorescence quenching of the semiconductor by the metal. Similar PL dynamical behaviors were also observed in an Au-CsPbBr₃ system (Roman et al., Nano Lett. 17, 5561 (2017)).

We'd like to emphasize that the lifetime of charge separated state is much longer than that of hot carriers in metal. In spite of an insignificant effect on PL dynamics, the charge separated state can be further harvested to realize the hot-carrier optoelectronic devices with high efficiency.

The relevant discussion has been included as “*In principle, multiple factors may contribute to PL dynamics including the emission enhancement caused by passivation of surface traps and the fluorescence quenching of the semiconductor by the metal, which is also dependent on the size of metallic nanoparticles. In this study, TRPL spectra in Ag–CsPbBr₃ hybrid NCs with Ag nanoparticles of ~ 5.5 nm and neat CsPbBr₃ NCs are nearly the same at the late stage (Figure 2F), implying that the presence of SPR has insignificant impact on the dynamics of interband electron-hole recombination of the CsPbBr₃ NCs at long time scale (> 200 ps)*” on Page 8 and “*PL quantum yield in neat nanocrystals is about 58±5% while that in the sample of hybrid nanocrystals we study is about 65±6%*” on Page 18 of the revised manuscript.

Comment 3. The timescales of the claimed hot electron injection don't make sense with the yields. If hot electron injection is 100 fs, and the lifetime of the Ag nanoparticle itself is 800 fs, then the simple quantum yield branching would give an injection yield of 88% (and 100fs is the upper limit). They measure a 69% amplitude ratio – why the loss in yield? I can imagine several possibilities (such as a particular energy threshold for hot electron injection), but they need to address all of these timescales in a more quantitative fashion.

Our Response: We thank the reviewer for pointing out the intuitive inconsistency between the efficiency and lifetime parameters. Indeed, in the nanocrystals with

ultrafast hot-electron injection (< 100 fs), the efficiency may be higher. As noted by the reviewer, conventional hot electron transfer occurs only in hybrid nanocrystals where the hot electrons in Ag are excited with occupation at levels higher than a threshold (energy offset + Schottky barrier). If the coupling between metal-semimetal is sufficiently strong, energy/electron transfer may be induced by plasmon transfer. In this case, only part of the excited nanocrystals undergoes the process of hot-electron transfer. In addition, due to sample heterogeneity of Ag-CsPbBr₃ NCs, energy transfer and electron transfer may occur in different samples. We have included a new Supplementary Note 2 for qualitative analysis. The additional discussions on this issue are given as “*In general, PIRET and HET are competing channels for plasmon-derived hot excitation transfer processes. Both channels are beneficial from the enhanced coupling between metal and semiconductor in Ag-CsPbBr₃ NCs. The coexistence of PIRET and HET processes is possibly related to sample heterogeneity. In some Ag-CsPbBr₃ NCs where residual AgBr at the Ag/CsPbBr₃ interface may disable the direct physical contact between metal and semiconductor,³⁴ PIRET may dominate the process of hot excitation transfer. In other Ag-CsPbBr₃ NCs with good metal-semiconductor contact, HET is likely to be the major channel responsible for the observed experiments*” on Page 16 of the revised manuscript.

Comment 4. The spectrum of perovskites at the band edge is a combination of bandgap renormalization and state-filling. Why is bandgap renormalization not considered when interpreting the signal near 525 nm?

Our Response: We thank the reviewer for reminding us the effect of bandgap renormalization. Indeed, the TA spectrum at the band edge is contributed by both the bandgap renormalization and state filling. In the weak excitation regime, the bandgap renormalization is insignificant and the TA signals induced by the two effects are linearly dependent on the excitation density for both neat CsPbBr₃ and hybrid Ag/CsPbBr₃ nanocrystals. In this case, the quantification procedure is still valid. We have included the discussion as “*which can be naturally assigned to the state filling and bandgap renormalization. In the weak pump regime, the signal amplitudes for both effects are proportional to the excited-state population near band edge*” on Page 8 in the revised manuscript.

Comment 5. The discussion of hot carrier multiplication on page 13 is purely speculative, with no experimental evidence. The carrier multiplication field is so fraught with arguments over experimental conditions, signal interpretation, etc, that such speculation should be treated very cautiously (especially given my concerns below about the reporting of experimental conditions)

Our Response: We agree with the reviewer that the hot carrier multiplication is a speculation at current stage. To avoid potential misunderstanding, we have eliminated the relevant discussion and removed the related supplementary figure in the revised version.

Additional comments: In addition to these conceptual questions, several aspects of the

data presentation are insufficient:

Comment 1. The pump power (in mJ/cm²) is never given for any experiment, nor is the OD of the sample. In addition, all experimental spectra are normalized, so the reader has no sense of the magnitude of the signals. Both are needed to accurately judge the presence of possible multiphoton effects (did they do the Poisson distribution calculation?)

Our Response: In the original manuscript, we describe the excitation density in terms of average absorbed photons per nanocrystal (~ 0.2). In this weak excitation regime, most excited nanocrystals absorb only one photon per nanocrystal considering the Poisson distribution. The OD of the sample at 500 nm is ~ 0.25 . The plotted data were recorded with a pump power of $\sim 3 \text{ uJ/cm}^2$. Under such condition, the multiphoton effect is negligible. In the Method, we included the information: “Cyclohexane solutions of samples were added to 1 mm-thick quartz cuvettes with an optical density of ~ 0.25 at 500 nm for optical measurements”, on Page 18. To release reviewer’s concern, we included the fluence-dependent data as Supplementary Figure 4 and relevant discussion in Supplementary Note 1 in the revised SI.

Comment 2. The pump power and an accurate calculation of excitations/particle is especially important when trying to compare the Ag/CsPbBr₃ vs CsPbBr₃ spectra at early times. Nothing in the manuscript says that the Ag/CsPbBr₃ wasn’t just pumped harder than the CsPbBr₃ sample, perhaps making more doubly-excited particles.

Our Response: In this work, the excitation density was kept at a weak level with average absorbed photons per nanocrystal of ~ 0.2 . Specifically, the comparable experiments on the samples of neat CsPbBr₃ and hybrid Ag-CsPbBr₃ were performed under the same conditions. We have explicitly clarified this point in the revised manuscript. To avoid potential problem, we have included the fluence-dependent data as the new Supplementary Figure 4 and relevant discussion in Supplementary Note 1 in the revised SI.

Comment 3. In Figure 3C, are the lines some kind of spline fit with the symbols showing the actual data? The symbols don’t seem to line up vertically, which I don’t understand. Since they use a monochromator and not an array CCD, I can’t assume that they have regularly spaced data (especially since in their time traces they show each data point with a symbol). The lines in 3C seem too smooth to be the data (in which case the symbols are labels), but perhaps their instrument is in fact that good.

Our Response: As reported in Methods, we recorded the data with an array CCD having 1024 pixels (S7030-1006, Hamamatsu). The TA spectrometer is a standard one as we described previously (e.g., Xu et al., JACS 138, 3761 (2016); Bin et al., Nature Commun. 7, 13651 (2016); Chen et al., JPCC 121, 12972 (2017)). The signal-to-noise ratio for $\Delta T/T$ is at the level of $\sim 5 \times 10^{-5}$. In the plot of original Figure 3C, we skipped some data points with different spacing rates to distinguish the spectra recorded at different time delays (particularly for the curves printed in black and white). To avoid potential ambiguity, we have changed the plot in Figure 3C in the revision manuscript.

Comment 4. Spectra are frequently normalized without any description of how (at what data or time point). There are a couple places where this is explained (such as to the late-time signal in Figure 4B) but it needs to be explicit everywhere.

Our Response: Follow the reviewer's suggestion, we have made the normalizations of spectra explicit everywhere in the revised manuscript.

Comment 5. Do the Ag/CsPbBr₃ dimers survive the transient absorption experiment? There should be before/after UV/Vis and TEM studies.

Our Response: As responded to the Comment 1 (in the conceptual/interpretation concerns) earlier, we have taken great care for the stability of the samples. During the TA measurements, we have monitored the UV/Vis absorption spectra to check the stability (Figure R5). The samples well survive for a round of experiments (Supplementary Figure 10). Following the reviewer's suggestion, we have further checked the stability of the samples with TEM (Figure R6). The morphology characterization suggests that the structure remains unchanged after the TA measurements. Nevertheless, TEM only characterizes the local morphology. The absorption spectra can better support the survival of the sample structure. We included the details of our experimental procedures on Page 18 and Supplementary Figure 10 (i.e. Figure R5) in the revised SI.

Figure R6. Typical TEM images of a sample of Ag-CsPbBr₃ nanocrystals before (A) and after (B) TA measurements.

Comment 6. What is the excitation wavelength for the TRPL studies? I didn't see this anywhere.

Our Response: We apologize for this omission. The excitation laser is a picosecond laser at 405 nm (LDH, Picoquant) for the TRPL studies. We have included the details on Page 19 in the revised manuscript.

Overall, we have addressed all the reviewers' comments/concerns, and made appropriate modifications in the revised manuscript and Supplemental Information. The revised manuscript has been substantially improved. We hope that the reviewers can now find the revised manuscript acceptable for publication in *Nature Communications*.

Reviewers' comments:

Reviewer #1 (Remarks to the Author):

The authors revised the manuscript carefully by adding appropriate discussion based on the additional experiments. Now the manuscript can be acceptable for the publication.

Reviewer #2 (Remarks to the Author):

The data interpretation is much better after revision. The authors claim that the energy transfer from the plasmonic metal to CsPbBr₃ via the mixed mechanisms: hot electron injection (HET) and plasmon-induced resonance energy transfer (PIRET), which is not surprising to achieve the energy transfer efficiency over 50%. Based on the data and the interpretation in the revised manuscript, the energy transfer efficiency over 50% is a normal value when both HET and PIRET take place. In addition, both HET and PIRET are well known energy transfer mechanisms. In short, this manuscript does not present any new mechanism, and the plasmonic metal-CsPbBr₃ hybrid system's performance fall into a regular range, which has dampened the reviewer's enthusiasm dramatically. In summary, the results and knowledge obtained in this manuscript do not warrant its publication in Nature Communications. However, I recommend it to be published somewhere else such as Scientific Reports.

Reviewer #3 (Remarks to the Author):

The authors have responded appropriately to all of my concerns with the photophysics and the presentation of the results. I will defer to the other reviewers as to the significance of the work.

Response to the reviewers' comments for the manuscript #NCOMMS-18-02805A-Z:

We thank the reviewers for their insightful reviews of our resubmitted manuscript. All reviewers agree that all the technical comments have been well addressed in the revised manuscript, and Reviewers #1 and #3 have recommended publication of our revised manuscript in *Nature Communications*. In the following, we mainly respond to the Reviewer #2's concerns.

Response to the comment by Reviewer #1:

Comment: The authors revised the manuscript carefully by adding appropriate discussion based on the additional experiments. Now the manuscript can be acceptable for the publication.

Our response: We are grateful that the reviewer has fully appreciated our responses to the comments raised in the previous reports and the significant improvements made in the revised manuscript. The reviewer now recommends our revised manuscript for publication in *Nature Communications*.

Response to the comments by Reviewer #2:

Comment: The data interpretation is much better after revision. The authors claim that the energy transfer from the plasmonic metal to CsPbBr₃ via the mixed mechanisms: hot electron injection (HET) and plasmon-induced resonance energy transfer (PIRET), which is not surprising to achieve the energy transfer efficiency over 50%. Based on the data and the interpretation in the revised manuscript, the energy transfer efficiency over 50% is a normal value when both HET and PIRET take place. In addition, both HET and PIRET are well known energy transfer mechanisms. In short, this manuscript does not present any new mechanism, and the plasmonic metal-CsPbBr₃ hybrid system's performance fall into a regular range, which has dampened the reviewer's enthusiasm dramatically. In summary, the results and knowledge obtained in this manuscript do not warrant its publication in Nature Communications. However, I recommend it to be published somewhere else such as Scientific Reports.

Our response: We are glad to see that our data interpretation has been well approved by the reviewer and there are no technical issues in our experimental demonstration.

Unfortunately, it seems that the reviewer has misunderstood and misinterpreted the reported efficiencies in the revised manuscript. The mentioned efficiency of 50% in the manuscript is solely for the hot-electron transfer (HET) process, rather than the *total efficiency* of plasmon-hot electron conversion (i.e., the energy transfer) as quoted in the reviewer's comment. As suggested by this reviewer in his/her last-round of review, we have evaluated the contribution of PIRET with an efficiency of 15±5 % in the plasmon-hot electron conversion process by the photoluminescence excitation spectroscopy. Nevertheless, the major channel is still the process of HET with the efficiency of 50±18% as evidenced by high photogeneration yield of the charge-separated states. The efficiency of 50% is perhaps *a normal value* or "in a

regular range” for PIRET (or total energy transfer). However, it is a remarkably high value for the process of HET which has *never* been achieved in *any heterodimer nanocrystal systems*. The efficiency of hot-electron transfer demonstrated in the current Ag-CsPbBr₃ nanocrystals is about twice of that measured in the Au-CdS systems (e.g., Wu *et al.*, *Science* **349**, 632 (2015)), which is also above the best values reported in other high performance systems with metal nanoparticles embedded within semiconductor films (such as Furube *et al.*, *JACS* **129**, 14852 (2007), Giugni *et al.*, *Nature Nanotech.* **8**, 845 (2013), Ratchford *et al.* *Nano Lett.* **17**, 6047 (2017), etc.).

Overall, we have demonstrated a very first ultrafast spectroscopic study on the plasmon-hot electron conversion in the metal-perovskite (Ag-CsPbBr₃) nanocrystal systems. While both processes of HET and PIRET are involved, HET makes the substantial contribution with a record high efficiency of $\sim 50\pm 18\%$, benefiting from the high density of states in perovskite semiconductors. This work highlights, for the first time, that the interface between metal and perovskite semiconductor is ideal for the efficient HET process. The superior electronic properties can be easily integrated in the rapidly developing device architectures with perovskite semiconductors, which are of great value for demonstrating highly efficient hot-carrier optoelectronic devices. Thus, we strongly believe that this manuscript, as approved by other reviewers, is of great significance for publication in *Nature Communications*.

To avoid potential misunderstanding, we have explicitly stated the efficiencies of hot-electron transfer and PIRET in the Abstract, Page 2 and Pages 13-14, as highlighted in different color, of the revised manuscript.

Response to the comment by Reviewer #3:

Comment: The authors have responded appropriately to all of my concerns with the photophysics and the presentation of the results. I will defer to the other reviewers as to the significance of the work.

Our response: We are very happy to see that the reviewer is fully satisfied with our Response and revisions of the manuscript to address his/her concerns.